# CONTRASTIVE TRAINING WITH MORE DATA

**Stephen Mander, Scott Piao and Hossein Rahmani**
School of Computing and Communications, Lancaster University, UK
{smander3,s.piao,h.rahmani}@lancaster.ac.uk

## ABSTRACT

This paper proposes a new method of contrastive training over multiple data points, focusing on the scaling issue present when using in-batch negatives. Our approach compares transformer training with dual encoders versus training with multiple encoders. Our method can provide a feasible approach to improve loss modelling as encoders scale.

## 1 MOTIVATION

We hypothesise that the efficacy of contrastive training correlates to the ratio between positive samples and in-batch negatives. This paper explores a way of increasing the logits by altering the approach used with additional input sources.

## 2 LINEAR APPROACHES FOR ADDITIONAL ENCODER STREAMS

Typical contrastive training uses two auto-encoders, but we explore instances of $n$ auto-encoders where $n > 2$. Consider the case, $n = 3$, where auto-encoder outputs are $X$, $Y$ and $Z$ respectively. We maintain the linear relation present in CLIP Radford et al. (2021) by iterating over pairs; $CosineSimilarity(XY)$, $CosineSimilarity(XZ)$, $CosineSimilarity(YZ)$. While some minor optimisations are available for this case, the ratio of in-batch negatives is not improved, and training time and performance is affected in a non-linear fashion. Additional modalities also present implementation issues around balancing learning rates between intra-modal parts of the network.

## 3 A CONSIDERATION OF COSINE SIMILARITY WITH ADDITIONAL DIMENSIONS

With two encoders (and their respective data inputs) the loss generated with the cosine similarity is modified for faster execution, to be the matrix multiplication of 2 fractions.

$$CosineSimilarity(a, b) = (\frac{a}{\sqrt{\sum a}}) . (\frac{b}{\sqrt{\sum b}}) \tag{1}$$

In this matrix multiplication, a transpose is used, and the operation is limited to just two dimensions. Two dimensions are fine with just two operands. With two encoders, the ratio of positive to negative samples within a batch, $B$, is simply $1 : B - 1$. For every additional encoder, the exponent of $B$ in the ratio of $1 : B - 1$ will increase by 1. For $n = 2$ vectors, $y_1.y_2$ is a good similarity metric; 2 negative samples return a positive similarity. Now consider the case of $n > 2$, the dot-product of $y_1.y_2.y_3....y_n$ ,and consider the vectors $a$, $b$ and $c$. $a.b.c$ is positive if $a$, $b$ and $c$ are positive, and it is negative if all are negative. This means that naturally maximizing $a.b.c$ can promote some values within $[a, b, c]$ to become negative.

## 4 CORRECTING LOSS

Cosine similarity allows a reduction in size as it is performed, matrix multiplications being relatively quick operations. By contrast, calculating a similarity $S$ as $S = 1 - variance$ requires more complex steps of computation. An initial observation may require extra arrays created for computation, such as size $B^n \times F \times n$, where $F$ is the feature space. Even with extensive work to reduce this footprint, it drastically reduces the achievable batch size for a finite amount of hardware.

## 5    EUCLIDEAN DISTANCE-BASED ERROR

An alternative to cosine similarity can be to define variance as the L2Norm of the distance ($D$) to the mean in each dimension. We denote the mean across an array, $x$ as $\bar{x}$. (Note, that for the following equations $x$ represents $n$ vectors). Thus,

$$S = 1 - L2Norm(D) \text{ where } D = x_i - \bar{x} \tag{2}$$

with $L2Norm(D)$ as $\sqrt{\sum_{i=0}^{n}(D^2)}$. However, the mean across all features is of shape $B^n \times F$. The following holds for when $x_i$ is a single feature in each encoder output, and when $x_i$ is the whole output of shape $B \times F$. We use formula (4) for calculating $1 - L2Norm(D)$, which is derived as follows:

$$S = 1 - \sqrt{\sum_{i=0}^{n}(x_i{}^2 + \bar{x}^2 - 2x_i\bar{x})} = 1 - \sqrt{\sum_{i=0}^{n}(x_i{}^2) + \sum_{i=0}^{n}(\bar{x}^2) - \sum_{i=0}^{n}(2x_i\bar{x})} = 1 - \sqrt{\sum_{i=0}^{n}(x_i{}^2) + \frac{(\sum_{i=0}^{n}x_i)^2}{n} - \sum_{i=0}^{n}(2x_i\bar{x})} \tag{3}$$

We can further convert the maths expression $-\sum_{i=0}^{n}(2x_i\bar{x})$ into $-2\bar{x}\sum_{i=0}^{n}x_i$, which can also be expressed as $-2\frac{(\sum_{i=0}^{n}x_i)^2}{n}$. Giving:

$$S = 1 - \sqrt{\sum_{i=0}^{n}(x_i^2) - \frac{(\sum_{i=0}^{n}x_i)^2}{n}} \tag{4}$$

Naturally, this means that much less computation is needed with $n$-dimensional arrays, substantially reducing the computational load.

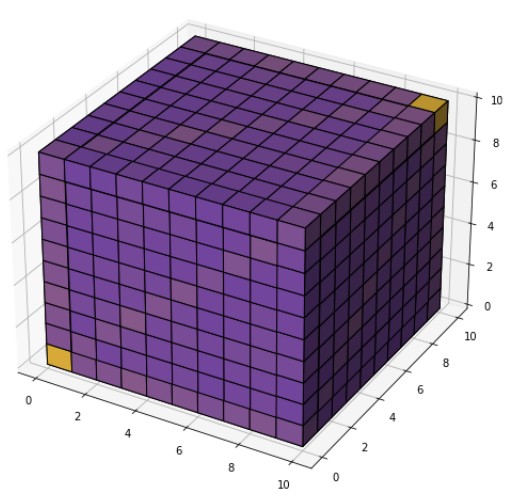

Figure 1: A plot of logits where $n = 3$ and $B = 10$.

As explained previously, the proposed approach is moving from logits of shape $B \times B$ to logits in the form $B^n$, as illustrated by Figure 1, where each value corresponds to items in each batch, $x_i$. The location $(i, j, k)$ in the cube is the variance of the set of features between inputs $[x_{0,i}, x_{1,j}, x_{2,k}]$. All logits are combined in the cube space, with maximum values on the highlighted diagonals being the target indexes for cross-entropy loss. (Code and results for comparison in n-dimensions can be found at https://github.com/st7ma784/ContrastiveTraining.)

## REFERENCES

Alec Radford, Jong Wook Kim, Chris Hallacy, Aditya Ramesh, Gabriel Goh, Sandhini Agarwal, Girish Sastry, Amanda Askell, Pamela Mishkin, Jack Clark, Gretchen Krueger, and Ilya Sutskever. Learning transferable visual models from natural language supervision, 2021. URL https://arxiv.org/abs/2103.00020.

