# OpenReview forum: "Contrastive Training with more data"
_ICLR.cc/2023/TinyPapers — Submitted to Tiny Papers @ ICLR 2023_

### Official Review · Reviewer_cCsE · 2023-03-29

**Confidence:** 3

**Summary Of Contributions:**

In this paper the authors are presenting a solution to the scaling issue presented in contrastive-training while using in-batch negatives. They claim with their loss function can help reduce the computational load for n-dimensional arrays (n>2).

**Rating:**

Needs Clarification (NC): a submission which does not meet the reviewing criteria and needs clarification for its described problem or solution

**Strengths And Weaknesses:**

Strength:
1. They presented an idea and a loss function for reducing computation for higher dimensional arrays.

Weakness:
1. The submission should have gone through proofreading and make it more clear and understandable for a reader that doesn’t work in the project with them.
2. When using symbols, equations, and formulas you have to make it more clear for the readers to read them.
3. It would be better to provide some direct comparison in a table or at least in a sentence for the claimed computational advantage of your approach.

Please look at the suggested changes for some specific comments.


**Suggested Changes:**

In the abstract:
1. “focusing on the scaling issue present when …” Perhaps change loss modelling → loss modeling (being mindful of being consistent with using American English OR British English).

In section 2:
1. Equation (1):
(1) When talking about a formula it is better to have it as a two sided equality (the result of this operation would be applied to what?)
Saying that a transposed is used: transpose of what?
(2) What is a and b here? This has to be included in the paper for clarity close to the equation definition.
(3) Making your paper more professional: avoid using words such as fine and just in “Two dimensions are fine with just two operands.”, I recommend rewriting your sentence.

2. In paragraph 3:
(1) Saying “dot-product” in ”the dot-product of y1 × y2 × y3....yn” and then using the cross-product symbol is confusing and should change to the appropriate product symbol.
(2) Again using symbols that you do not define doesn’t help for the reader to see what you are talking about: a,b,c, B, …

In section 3:
1. In the first sentence “:” is not appropriate and not necessary, you should rewrite the sentence.
2. Again b and B^n; x_i, x(hat) in equation (3) are not clearly defined.
3. Is F and f  the same? Confusing
4. Summation symbol doesn’t have limits, as well as square root in the text.
5. Equation (4), (5), (6), and (7) all say the same thing. I think all readers can understand the equation (6) and/or (7) with one sentence explanation. The saved space could have been used for elaborating more at the begging of the paper and navigating the reader through what problem you are trying to solve.
6. Figure 1 can be colored better to show the difference more significantly.

---

### Official Review · Reviewer_We8a · 2023-03-30

**Confidence:** 3

**Summary Of Contributions:**

 This paper proposes a new contrastive training method with multiple encoders that addresses the scaling issue of typical contrastive training methods.

**Rating:**

Needs Clarification (NC): a submission which does not meet the reviewing criteria and needs clarification for its described problem or solution

**Strengths And Weaknesses:**

Weakness:
1. The paper motivation is not clear.
2. The result of the paper is not clear.

**Suggested Changes:**

Suggested Changes:
1. Clearly define the motivation of your approach or define the problem concretely.

---

### Author Response · Authors · 2023-06-30
**Consent To Archive:**

 Yes, I hereby give consent to have my paper archived.

---

### Meta-Review · Area_Chair_QFt5 · 2023-04-05

**Recommendation:** Invite to revise
**Confidence:** 5

**Metareview:**

This study introduces a novel multi-encoder contrastive learning approach that tackles the scalability challenges inherent in conventional contrastive training methods. The authors present a solution that employs in-batch negatives to address the scaling issue found in contrastive training techniques. They assert that their loss function can effectively decrease the computational burden for n-dimensional arrays (n>2). However, the paper has significant shortcomings in terms of clarity, presentation, and proofreading.

**Summary:**

This study introduces a novel multi-encoder contrastive learning approach that tackles the scalability challenges.

**Reason For Not Giving A Higher Recommendation:**

This paper has lot of issues with unclear explanations of symbols, equations, and formulas, as well as a lack of direct comparisons to demonstrate the computational advantages of the proposed approach. As a result, the paper is challenging to comprehend. The reviewers provided detailed suggestions for changes, including rewriting certain sections, clarifying equations, and refining the presentation for better reader comprehension. Addressing these concerns would significantly enhance the quality and impact of the paper.



**Reason For Not Giving A Lower Recommendation:**

N/A

---

### Decision · Program_Chairs · 2023-04-10

Revision accepted; invite to archive